# Correlation between Handgrip Strength and Depression in Older Adults—A Systematic Review and a Meta-Analysis

**DOI:** 10.3390/ijerph18094823

**Published:** 2021-04-30

**Authors:** Ewa Zasadzka, Anna Pieczyńska, Tomasz Trzmiel, Paweł Kleka, Mariola Pawlaczyk

**Affiliations:** 1Department of Occupational Therapy, Poznan University of Medical Sciences, 60-781 Poznań, Poland; apieczynska@ump.edu.pl (A.P.); ttrzmiel@ump.edu.pl (T.T.); 2Institute of Psychology, Adam Mickiewicz University, 60-589 Poznań, Poland; pawel.kleka@amu.edu.pl; 3Department and Division of Practical Cosmetology and Skin Diseases Prophylaxis, Poznan University of Medical Sciences, 60-623 Poznań, Poland; mariolapawlaczyk@ump.edu.pl

**Keywords:** handgrip strength, depression, older adults, meta-analysis, systematic review

## Abstract

Background: Depression remains an important health problem among older adults and it may be correlated with the deterioration of physical fitness, whose chief indicator is hand grip strength (HGS). The aim of the study was to investigate the relationship between depression and HGS among older populations using the available literature. Methods: PubMed, Web of Science and Science Direct databases were searched. The inclusion criteria were as follows: written in English and published after 2009, subject age: ≥60 years, HGS measured using a hand dynamometer, assessment of the depressive symptoms using a validated tool. The following articles were excluded: studies conducted among institutionalized subjects and/or populations with a specific disease. Results: The total combined effect of 33 results presented in 16 studies included in the meta-analysis, converted to the correlation coefficient, was OEr = −0.148 (SE = 0.030, 95%CI: −0.206–−0.091), indicating a weak, negative correlation between HGS and depressive symptoms. Conclusions: The review of the literature and the meta-analysis demonstrated a relationship between low muscle strength and intensified depressive symptoms in older populations. Bearing in mind that depression is often unrecognized or underdiagnosed among older patients, lowered muscle strength should be an important sign for physicians and an incentive to screen them for depression.

## 1. Introduction

Depression remains an important health problem among older adults, resulting in lower cognitive ability and quality of life [1]. According to the World Health Organization (WHO), approximately 7% of the general older population were estimated to suffer from unipolar depression [2]. Depressive disorders in older age result from the accumulation of various factors, chief among them somatic diseases, stressful life events (e.g., loss of partner), loneliness, social isolation, unfavorable social attitudes towards older people, declined cognitive function, malnutrition, and polypharmacy. In many cases, depression may be the consequence of diseases which are typical for older populations, e.g., arteriosclerosis, hypertension, arrhythmia, diabetes, Parkinson’s disease, or osteoporosis.

According to a number of studies, as many as 80% of older people are affected by at least one somatic condition [3,4], and 69% patients with depression report health complaints which are solely somatic in nature: pain, feeling of heaviness, fatigue, disturbed sleep, appetite loss, impaired gait and functional performance [5]. Depression may negatively affect the treatment of chronic diseases, thus worsening the prognosis, due to appetite loss, reluctance to comply with medication regimen, avoidance of social contacts, increasing social isolation, and unwillingness to engage in any forms of physical activity, including rehabilitation [6].

On the other hand, an equally impressive amount of data demonstrated that higher cardiorespiratory fitness and regular aerobic exercise prevent the development of depression among older individuals [7]. According to the WHO guidelines on minimal physical activity for older populations, 150 min/week of moderate-intensity (brisk marching, cycling, swimming) or 75 min/week of vigorous-intensity (jogging) aerobic exercises are recommended [8]. Physical activity contributes to maintaining physical and mental health and exerts beneficial effects on the treatment of depression.

According to the available literature, a relationship between low muscle strength and cognitive function impairment and the risk for developing neurodegenerative diseases such as Alzheimer’s or Parkinson’s disease has been confirmed [9]. Furthermore, decreased muscle strength may correlate with depressive symptoms among older subjects [10]. Handgrip strength (HGS) test is a simple method of measuring the extent of muscle power loss in clinical practice. The European Working Group on Sarcopenia in Older People 2 (EWGSOP 2) treats HGS as an index of global muscle power and as one of the criteria for the diagnosis of sarcopenia [11,12]. Sarcopenia, which may be defined as loss of skeletal muscle combined with decreased muscle strength and/or physical performance, is an increasingly common problem of older populations globally [12]. It is associated with higher risk for mortality, deteriorated functional performance, falls and hospitalization, as well as increased risk for depression in that age group [13]. In the elderly, the loss of muscle strength can also be caused by dynapenia, but there is no loss of muscle mass [14].

The impact of depressive symptoms on HGS has been investigated for various age groups. In a systematic review [10], summarized the studies on the relationship between muscle strength and depressive symptoms among middle-aged and older subjects, and emphasized the fact that lower risk for the development of depressive symptoms correlated with higher muscle strength.

The aim of the study was to investigate the relationship between depression and HGS among older populations using the available literature.

## 2. Materials and Methods

The review of the literature and the meta-analysis were conducted in accordance with the Preferred Reporting Items for Systematic Reviews and Meta-Analyses (PRISMA) guidelines [15]. The procedures (search strategy, inclusion/exclusion criteria, and data extraction) were established and included in the protocol.

### 2.1. Search Strategy

PubMed, Web of Science and Science Direct databases were searched. An algorithm with key words (‘handgrip strength’ AND ‘depression’ AND ‘older’ OR ‘elderly’) was used to identify the publications. Additionally, the reference sections of the included articles were manually inspected to identify additional records. Two authors (EZ and TT) conducted their independent searches between 1 January 2010 and 17 January 2020. Only articles published in English were taken into consideration.

### 2.2. Inclusion and Exclusion Criteria

The inclusion criteria for the reports were as follows: (a) written in English and published after 2009, (b) subject age: ≥60 years, (c) HGS measured using a hand dynamometer, (d) assessment of the depressive symptoms using a validated tool.

The following articles were excluded: studies conducted among (a) institutionalized subjects and/or (b) populations with a specific disease, e.g., malignancy or carpal tunnel syndrome.

### 2.3. Data Collection and Analysis

All articles were independently analyzed by two authors (E.Z. and T.T.) to remove duplicates. The results were reviewed, and full versions were checked for compliance with the inclusion and exclusion criteria. The following data were extracted from each study: first author, year of publication, study population characteristics, study design, inclusion/exclusion criteria, method of assessing HGS and depressive symptoms, assessment of the outcome, and results. Next, methodological quality standards and risk of bias (using the Newcastle–Ottawa Scale (NOS) adapted for cross sectional studies were investigated [16]. NOS assesses three parameters (selection, comparability, and outcome) on eight items, with the maximum score of 10 ints. Studies with the score of <5 points (<5NOS) are considered to have high risk of bias. The questionnaire was adapted for the present study in the following way: (a) in the Selection section (Ascertainment of exposure), 1 point was awarded if the measuring tool was presented but the method was either not described or not compliant with the American Society of Hand Therapists (ASHT) guidelines [17], and 2 points were awarded if the tool was presented and the method was compliant with the ASHT guidelines; (b) in the Outcome section (Assessment of outcome), 2 points were awarded if HGS and depressive symptoms were assessed by two independent blind assessors. Methodological quality assessment was conducted independently by two authors (E.Z. and T.T.). All conflicts were discussed and if consensus could not be reached, the third author (A.P.) had the deciding vote.

### 2.4. Measures of Effect Sizes

The collected results had been presented in various ways and had to be converted to the correlation coefficients. The highest number of relationships between depressive symptoms and HGS was expressed as beta coefficients, obtained by the linear regression analysis (*n* = 12). The linear regression method allowed to control for the effects of mediator variables such as age, sex, education, and others, and that is why these values represent a partial effect of HGS on the level of depression. We converted these values to r_eq using the formula $\sqrt{t^{2}/t^{2}+N-2}$. Odds ratio (OR) calculated for cross tables was the next most frequent (*n* = 9) measure. The tables were prepared based on the threshold values for the number of depressive symptoms and according to the generally accepted thresholds for HGS or the thresholds calculated using the Receiver Operating Curve (ROC). We converted these values to r_eq using first the formula $d=ln\left(OR\right)\cdot \sqrt{3}/\pi$. and then the formula $d/\sqrt{d^{2}+4}$. Other indices of the effect size expressed as r-equivalent were calculated using mean values in the study and control groups (*n* = 4), correlation (*n* = 2), contingency coefficient (*n* = 2, here we also used means and standard deviations), standardized beta value (*n* = 1, similarly to stated earlier formula for betas) and d Cohen (*n* = 1, similarly to stated earlier the second formula for converting OR). In two cases, the identified texts did not report the effect size but only the *p*-values. Given the sample size and assuming test power of the analyzed studies (1 − β = 0.80), the minimal effect was calculated (we have calculated this with the calculator on the website https://www.campbellcollaboration.org/escalc/html/EffectSizeCalculator-R7.php). The total effect for the relationship between depressive symptoms and HGS was not described in six studies—the results were presented for groups with varying levels of depression and different sex groups, but it did not differentiate the total effect (Z = 0.715, *p* = 0.475).

Most studies did not present sex-stratified data as they used statistical measures which allowed to control for sex, but also other variables. Nevertheless, nine results were related to male subjects only, also nine to female subjects only, and eleven to male and female participants. No statistically significant differences in the effect size were found between these groups (F(2, 15.6) = 0.741, *p* = 0.493).

### 2.5. Statistical Analysis

Statistical analysis was conducted using the Jamovi software (2020), Jamovi project, Sydney, Australia, with the Viechtbauer metaphor package [18]. Publication bias was assessed visually by funnel plots and statistically by Egger’s test. The τ index, calculated using the Maximum-Likelihood method, was τ = 0.147 and was statistically significantly different from zero (Q(32) = 280.4, *p* < 0.001) and revealed high heterogeneity of the results. The I^2^ index was 92.3%, which justifies the use of the random effects model in the meta-analysis.

## 3. Results

Out of 473 records identified during the database search and 21 additional records identified after reference list search, a total of 16 entries [19,20,21,22,23,24,25,26,27,28,29,30,31,32,33,34] were included in the study. The flow-chart of the search process is presented in Figure 1.

The methodological quality of all included studies was sufficient (Table 1). The highest (9 points–NOS) quality was attributed to the studies by Vasconcelos et al. [32] and Han et al. [24] and the lowest (5 points–NOS) by Laredo-Aguilera et al. [27] and Olgun Yazar and Yazar [34]. The Sample size section was the weakest point of the studies—only three articles were awarded 1 point [20,24,32]. The strongest points proved to be the Representativeness of the sample—only Laredo-Aguilera et al. [27] did not receive a point based on design and analysis—Seino et al. [29] and Olgun Yazar and Yazar [34] received 1 out of 2 points. 

Overall, the results of 19,637 subjects (male and female, aged ≥ 60years) were presented in the meta-analysis. The GDS-15 scale was the most commonly used tool for evaluating depression in older patients (8 studies) [19,23,25,27,28,32,33,34], followed by Patient Health Questionnaire-9 (3 studies) [20,24,30], the Center for Epidemiologic Studies Depression Scale (3 studies) [21,22,29], Brief Symptom Inventory (1 study) [31], and the Hospital Anxiety and Depression Scale (1 study) [26]. HGS was measured using Takei (5 studies) [24,27,29,30,31], Jamar (5 studies) [25,26,32,33,34], EH 101 Camry (1 study) [19], Smedley (1 study) [22] and Hand Grip Meter 6103, Tanita (1 study) [28] hand dynamometers. Three studies failed to include the names of the dynamometer models [20,21,23]. The results were presented as kilograms or kilogram-force. The study characteristics are presented in Table 2.

The total combined effect of 33 results presented in 16 studies (Table 3) included in the meta-analysis, converted to the correlation coefficient, was OEr = −0.148 (SE = 0.030, 95% CI: −0.206–−0.91), indicating a weak, negative correlation between HGS and depression (Figure 2).

### 3.1. Publication Bias Analysis

Potential publication bias was tested via funnel plot (Figure 3). Egger’s test (bias = 0.150, *p* = 0.881) and Kendall’s τ coefficient (τ = −0.008, *p* = 0.951) showed no publication bias.

### 3.2. Sensitivity Analysis

The Fail-Safe N coefficient, according to the Rosenberg algorithm, indicates that a 2932 of texts with null effect would need to be included in the selection for the total score to reach zero. Sequence analysis (Figure 4) of the results included in the meta-analysis confirmed a statistically significant correlation between depression and HGS in the study population. Based on the data, the chance for the effect in the population is 81-fold higher that the chance for null effect.

## 4. Discussion

This systematic review and meta-analysis of 16 studies involving 19,637 individuals aged ≥ 60 revealed a weak negative correlation between HGS and depression. The GDS-15 scale was the most commonly used tool for evaluating depression in the analyzed publications. According to Friedman et al. [35] GDS has robust internal reliability, construct validity, and operational characteristics for the screening of community-dwelling, cognitively intact older adults. Its usefulness for the evaluation of depression in older subjects has been demonstrated by numerous reports [36,37]. HGS was most often measured using the Takei or Jamar hand dynamometers. Studies show that both measuring devices are successfully used to assess HGS [38,39], but the scores may differ due to different shapes of the handles. The devices and the methods of measuring HGS should be made uniform. The use of standardized measuring tools for assessing HGS and depressive symptoms, in well-matched groups of older people, will allow us to achieve reliable results [40].

In recent years, HGS has been perceived as a reliable indicator of the whole-body muscle strength, physical function, and health status, as well as a predictor of the length of hospitalization and even mortality for older populations, and has been investigated by a number of authors [41,42,43]. Additionally, HGS is a diagnostic criterion for sarcopenia [11,12]. Thus, as many as four publications on the matter were found during our search and included in the analysis [19,21,22,34]. Olgun Yazar and Yazar [34] and Wang et al. [19] concluded that sarcopenia was more common in older people with depression and depressive symptoms and that HGS was lower in those individuals. In turn, M. Hamer et al. [22] demonstrated that reduced grip strength was associated with higher risk of depressive symptoms in obese participants only. The possible link between weight status and HGS was investigated by Smith et al. [30], who found that obese subjects with moderate to severe depressive symptoms had lower HGS. Brown et al. [21] who analyzed frailty and depression in older adults, demonstrated that older people with symptoms of depression had lower HGS.

HGS test is also used to assess physical and functional fitness in older people. Vasconcelos et al. [32] investigated the cut-off points of HGS to identify mobility limitation and calculated the following values: ≤17.4 kg for women and ≤25.8 kg for men in community-dwelling settings. HGS below these values was characteristic for the group with muscles weakness, who presented with depressive symptoms significantly more often.

Chen et al. [26] investigated the relationship between HGS and duration of sleep in older people, with depressive symptoms as an additional variable. These authors found lower HGS in subjects with depressive symptoms. Laredo-Aguilera et al. [27] also analyzed the link between HGS and quality of sleep in older populations, considering their mood and psychical functioning. They found a correlation between HGS and vigor, depression, insomnia, and sleep quality. Pearson correlation analysis adjusted for age showed significant correlations between HGS and depression (r = 0.379, *p* = 0.021).

HGS allows to predict mortality among older populations, especially the ‘oldest’ old [25,31]. Additionally, late-life depression could be associated with high risk mortality, as reported by Hamer et al. [23]. These authors suggest that the relationship may be the result of lack of physical activity and poor physical function, measured with HDS. Depressed patients had lower HGS scores as compared to non-depressed individuals.

Depression in older people is associated with more functional and cognitive impairment than in younger adults [44]. Holmquist et al. [33] investigated the risk factors for depression in the context of functional performance. They concluded that high risk for depressive symptoms in older people was associated with low levels of functional performance (including HGS) combined with low physical activity.

Depression is one of the multiple geriatric syndromes [29,45]. Seino et al. [29] analyzed different measures of physical performance in order to determine the indicators of geriatric syndromes. They demonstrated that lower HGS was found in all geriatric syndromes (including depression) apart from urinary incontinence and malnourishment.

In our meta-analysis, we also took into consideration research of Korean authors, who studied a relationship between blood cadmium levels and HGS and depressive symptoms. Higher HGS values were associated with a lower number of depressive symptoms, assessed with the Korean Version of the GDS-short form [28].

Out of the 16 texts which were deemed eligible for the meta-analysis, only two aimed to establish the relationship between HGS and depression [20,24]. Han et al. [24] investigated that link in the context of socioeconomic status of the older subjects. These authors demonstrated a strong relationship between low HGS and intensified depression in socioeconomically deprived older people. Brooks et al. [20] concluded that reduced levels of combined HGS are independently associated with depression among U.S. adults aged 60 years and older.

## 5. Limitation

The fact that the modified NOS scale was used to analyze the quality of the studies, due to the lack of a more adequate tool to evaluate cross-studies, was a certain limitation of our study.

## 6. Conclusions

To sum up, the review of the literature and the meta-analysis demonstrated a relationship between low muscle strength measured with the HGS test and intensified depressive symptoms in older population, even though they were merely the additionally analyzed variables in the vast majority of the included texts.

Bearing in mind that depression is unrecognized or underdiagnosed in approximately 16% of the older patients [46], lowered muscle strength reported or found in older subjects should be an important sign for physicians and physiotherapists who work with older people and an incentive to screen them for depression, especially among older adults.

The mechanism linking HGS with depressive syndromes remain to be fully investigated. Their interdependence is a complex matter, indicating a strong two-way interconnection and the need for further studies to elucidate the matter.

## Figures and Tables

**Figure 1 ijerph-18-04823-f001:**
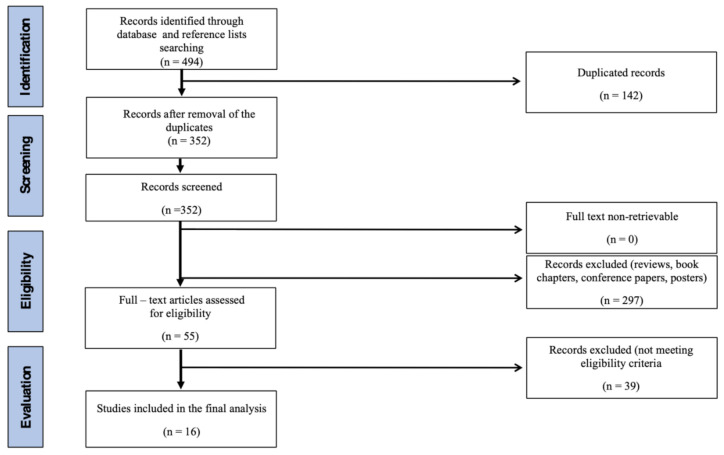
Flowchart showing how the reviewed articles were identified and selected.

**Figure 2 ijerph-18-04823-f002:**
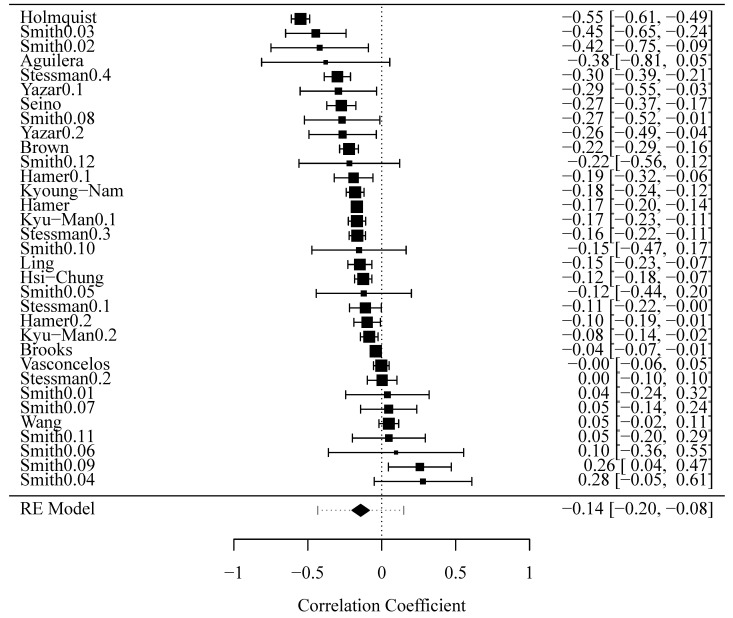
Forest plot of effect sizes (squares) with the overall result (diamond).

**Figure 3 ijerph-18-04823-f003:**
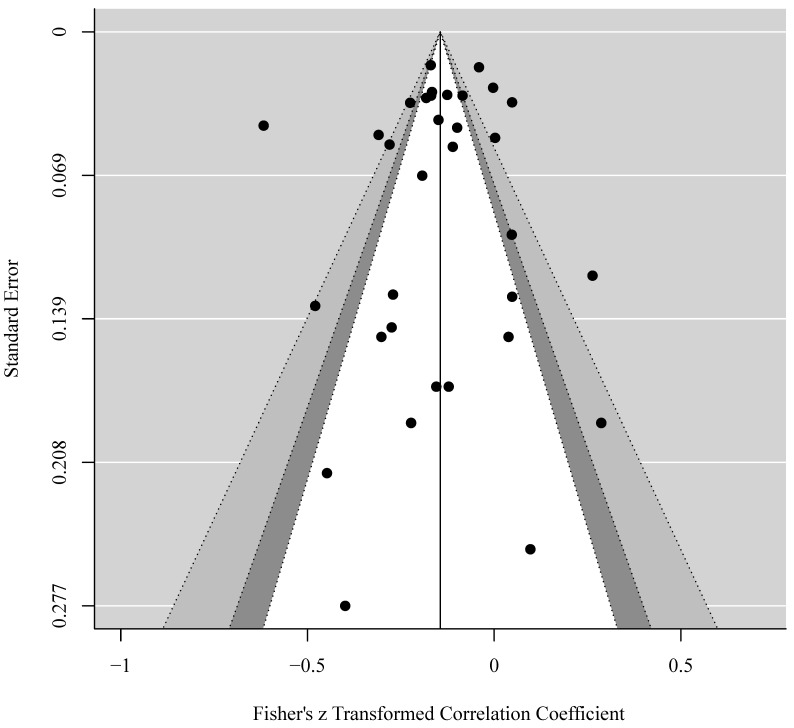
Funnel plot of Fisher’s z transformed correlation coefficient versus standard error of measurement.

**Figure 4 ijerph-18-04823-f004:**
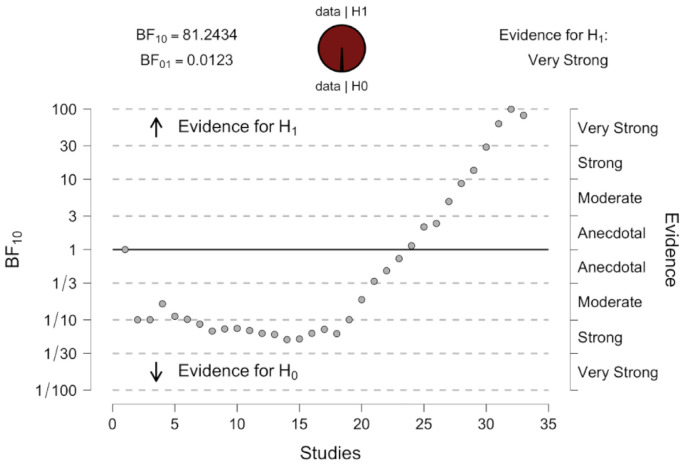
Sequential analysis of effects and Bayes factors for H1 against H0.

**Table 1 ijerph-18-04823-t001:** Evaluation of the methodological quality of each study using the Newcastle–Ottawa Scale.

Reference	Selection					Comparability		Outcome			Total Score
	Representativeness of the Sample	Sample Size	Non- Respondents	Ascertainment of Exposure	Score	Based on Design and Analysis	Score	Assessment of the Outcome	Statistical Test	Score	
Wang 2018 [19]	*	-	*	**	****	**	**	*	*	**	8 *
Brooks 2018 [20]	*	*	*	**	*****	**	**	*	-	*	8 *
Brown 2012 [21]	*	-	*	**	****	**	**	*	*	**	8 *
Hammer 2015 [22]	*	-	*	**	****	**	**	*	*	**	8 *
Hammer 2011 [23]	*	-	-	*	**	**	**	*	*	**	6 *
Han 2019 [24]	*	*	*	**	*****	**	**	*	*	**	9 *
Ling 2010 [25]	*	-	-	**	***	**	**	*	*	**	7 *
Chen 2017 [26]	*	-	-	**	***	**	**	*	*	**	7 *
Laredo-Aguilera 2019 [27]	-	-	*	*	**	**	**	*	-	*	5 *
Kim 2016 [28]	*	-	*	**	****	**	**	*	*	**	8 *
Seino 2013 [29]	*	-	*	*	***	*	*	*	*	**	6 *
Smith 2018 [30]	*	-	*	*	***	**	**	*	*	**	7 *
Stessman 2017 [31]	*	-	*	**	****	**	**	*	*	**	8 *
Vasconcelos 2016 [32]	*	*	*	**	*****	**	**	*	*	**	9 *
Holmquist 2017 [33]	*	-	-	**	***	**	**	*	-	*	6 *
Yazar 2019 [34]	*	-	-	**	***	*	*	*	-	*	5 *

*, **, ***—the number of points awarded according to the NOS scale key.

**Table 2 ijerph-18-04823-t002:** Study characteristics and the relationships between handgrip strength and depression.

Author and Year of Publication	Participants	Type of Dynamometer; Unit	Depression Scale	Inclusion/Exclusion Criteria	Relationships between HGS and Depressive Symptoms
Vasconcelos 2016 [32]	1374 subjects, women and men, age 73.4 ± 6.4	Jamar; kgf	GDS-15	Inclusion criteria: age ≥65 years, living in the community in the urban areas of Brazil. Exclusion criteria: score of <17 in the Mini Mental State Examination-MMSE), inability to walk, being bedridden, inability to hold a handheld dynamometer due to pain or hand deformities, hemiparesis caused by a stroke, severe Parkinson’s disease, cancer treatment (except for skin cancer) and terminal illness.	Incidence of depressive symptoms in the group with MW-muscle weakness (HGS ≤17.4 kgf for women and ≤25.8 kgf for men)-16.8% and NONE = no muscle weakness and no mobility limitation-11.4%
Smith 2018 [30]	587 subjects, women and men, age 69.2 ± *n*/d	Takei Digital Grip Strength Dynamometer; kg	PHQ-9	Inclusion criterion: age ≥ 60 years	Relationship between depressive symptoms and HGS by weight status from multiple linear regression models among adults aged ≥ 60 years
Wang 2018 [19]	865 subjects, women and men, age 68.68 ± 6.46	EH 101, Camry, Zhongshan, China; kg	GDS-15	Inclusion criterion: age ≥60 years Exclusion criteria: reported the presence of disease (including hyperthyroidism, hypothyroidism, or chronic heart or renal failure), physical disabilities (e.g., loss of a hand, foot, or multiple limbs), implanted electronic devices or orthopedic metal implantations, and those taking prescribed medications which could affect body composition (e.g., long-term systemic corticosteroids), severe hearing or eye problems, missing data	Comparison of HGS in the group with no depression = 30.1 ± 8.9 kg and with depression 29.2 ± 9.4 kg
Laredo-Aguilera 2019 [27]	16 subjects, only women, age 72.29 ± 5.21	TKK 5101 Grip D, Takey, Tokyo, Japan; kg	GDS-15	Inclusion criteria: not institutionalized; active women and above the age of 65; not suffering from mental and/or intellectual disorders; free of cardiovascular and neuromuscular disorders; considered physically independent according to the Spanish version of the Barthel Index. Exclusion criteria: participation in other training programs; artificial prosthesis; any disease requiring daily intake of drugs affecting the athletic performance, in order to avoid any influence on fitness measures; any disease that contraindicated the exercise program; any symptom that a medical professional deemed as warranting exclusion.	Pearson correlation analysis adjusted for age showed significant correlations between HGS and depression (r = 0.379, *p* = 0.021)
Ling 2010 [25]	484 subjects, women and men, age 85–89	Jamar; kg	GDS-15	Data from the prospective population-based Lei-den 85-plus study, which involved all 85-year-old inhabitants of Leiden, The Netherlands. No selection criteria had been imposed for health status or demographic characteristics.	Lower HGS was significantly associated (*p* < 0.001) with higher GDS scores
Stessman 2017 [31]	2304 subjects, women and men, age 70–90	5001 Grip- A, Takei, Niigata City, Japan; kg	BSI	Inclusion criterion: aged 70 to 71 at baseline in 1990–1991.	Relationship between HGS and depression measured at the ages of 70, 78, 85 and 90 years
Seino 2013 [29]	340 subjects, only women, age 80.0 ± 4.6	TKK 5401 Grip D, Takey, Tokyo, Japan; kg	CES-D	Inclusion criteria: female, aged ≥ 75 years Exclusion criteria: required a cane or other walking device, or their physical performance could not be measured by a standard method, e.g., they needed physical support from another person; could not understand the instructions of performance tests and questionnaires; or had data missing from their performance tests and geriatric syndromes.	Relationships between HGS and depressive symptoms expressed as odds ratio: adjusted for age, stroke, hypertension, diabetes mellitus, heart disease, respiratory disease, osteoporosis, dyslipidemia, low back pain and knee pain = 2.8 (1.2–6.3)
Olgun Yazar and Yazar 2019 [34]	281 subjects, women and men, age W/M: control 72.40 ± 6.46/72.26 ± 6.520 experimental 75.84 ± 6.98/ 74.14 ± 7.92	Jamar; kg	GDS-15	Inclusion criteria: normal neurological examination, without known chronic disease history apart from hypertension or chronic medication use, with no loss of more than 10% body weight within the last 6 months, and with HDRS score below 7 (control 1) or GDS score below 10 (control 2). GD patients additionally abided by the same exclusion criteria as healthy volunteers, apart from using at least one antidepressant. Patients with pacemaker or any implant and those with diseases severely affecting mobility (cerebrovascular events causing confinement to bed, advanced muscle disease, hip dislocations, decompensated heart failure, acute and chronic renal failure with fluid load, etc.) were not included in the study.	Mean HGS in the group of older people without depression 21.70 ± 5.478 and with depression 18.71 ± 5.4
Chen 2017 [26]	1081 subjects, women and men, age 76.3 ± 6.1	Jamar; kg	HADS	Inclusion criteria: aged ≥ 65 years and residence in Yilan City. Individuals who failed to provide a past medical history, who could not complete the interview, or who were unable to cooperate regarding the collection of anthropometric data because of physical disability or compromised cognitive function were excluded.	Individuals with depressive symptoms demonstrated weaker HGS (*p* < 0.001)
Kim 2016 [28]	983 subjects, women and men, age 75.2 ± 6.0	Hand Grip Meter 6103, Tanita, Tokyo, Japan; kg	GDS-15	Inclusion criteria: age ≥ 60 years and the ability to communicate with and follow instructions from the survey staff. Exclusion criteria: lack of information on blood cadmium concentration handgrip strength, anthropometric measurement and alcohol consumption,	Relationship between GDS score and HGS using linear mixed models. Adjusted for age, sex, city of residence, monthly income, education level, smoking status, pack-years of smoking, passive smoking status, alcohol drinking, moderate physical activity, weight, height, and comorbidity status. Right hand: beta −0.18; CI 95% −0.23, −0.13 Left hand: beta −0.15; CI 95% −0.20, −0.10
Brooks 2018 [20]	3421 subjects, women and men, age 69.9 ± 6.9	-; kg	PHQ-9	Exclusion criteria: age <60 years, missing data.	Linear Regression Models for Depressive Symptoms by HGS Outcome −0.19 ± 0.08
Brown 2012 [21]	854 subjects, women and men, age 75 ± *n*/d	-; kgf siła	CES-D	Inclusion criterion: 75 years of age at the time of the evaluation. Exclusion criteria: missing baseline depression data or missing data on all of the four frailty characteristics.	Mean HGS in the non-depressed group: 35.28 (12.57) kgf and in the mildly depressed group: 29.69 (10.20) kgf and in the depressed group: 30.07 (11.75) kgf
Hamer 2011 [23]	679 subjects, women and men, age: non-depressed 75.7 ± 7.1 depressed 79.0 ± 7.7	hand-held dynamometer; kg	GDS-15	Inclusion criterion: aged ≥ 65 years	Mean HGS in the non-depressed group: men 33.84 ± 10.6 kg, women 19.12 ± 10.0 kg and in the depressed group: men 29.74 ± 10.5 kg, women 17.11 ± 10.6 kg
Hamer 2015 [22]	3862 subjects, women and men, age 64.6 ± 8.3	Smedley hand-held dynamometer, Stoelting, USA; kg	CES-D	Inclusion criteria: absence of depression at baseline and availability of exposure, outcome and covariate data. For the key exposure measure, grip strength, there were no upper age limits Respondents were excluded if they had swelling or inflammation, severe pain or a recent injury or surgery to the hand in the preceding6 months.	Mean HGS in the non-depressed group: 31.2 ± 11.2 kg and in the depressed group: 27.4 ± 10.5 kg
Han 2019 [24]	1056 subjects, women and men, age 69.55 ± 6.25	THH 5401 Takei, Tokyo, Japan	PHQ-9	Inclusion criteria: adults aged ≥ 60 years with all of the following conditions: reliable data from a handgrip dynamometer; responded to the questionnaires regarding depressive symptoms; no missing data regarding sociodemographic and health-related variables.	In logistic regression analysis, older adults in the lowest HGS tertile were more likely to have depressive symptoms compared to those in the highest HGS tertile (adjusted odds ratio [aOR] = 1.85, 95% confidence interval [CI] = 1.25–2.74).
Holmquist 2017 [33]	490 subjects, women and men, age 70 ± 0	Jamar, kg	GDS-15	Residency in Umeå, Sweden was the only criterion for inclusion. Exclusion criteria: none	The present study provided a potential high-risk profile for depressive symptoms among elderly community-dwelling individuals, which included low levels functional performance (including HGS) combined with low levels of physical activity.

HGS—Handgrip Strength, GDS—Geriatric Depression Scale, PHQ-9—Patient Health Questionnaire-9, BSI—The Brief Symptom Inventory, CES-D—Center for Epidemiological Studies Depression Scale, HADS—Hospital Anxiety and Depression Scale.

**Table 3 ijerph-18-04823-t003:** Statistical presentation of the relationship between HGS and depression in the analyzed articles.

Author and Year of Publication	Group Size	Sex	Depression Scale	The Result Presented in the Work		The Size of the Effect Expressed in r_eq_
Vasconcelos 2016 [32]	1374	both	GDS-15	OR	0.99	−0.003
Smith 2018 [30]	49	M	PHQ-9	beta	0.34	0.039
	25	M		beta	−3.72	−0.420
	60	M		beta	−4.12	−0.446
	31	M		beta	2.01	0.279
	37	M		beta	−1.25	−0.121
	19	M		beta	1.45	0.100
	107	F		beta	0.31	0.047
	52	F		beta	−1.83	−0.268
	75	F		beta	0.86	0.258
	37	F		beta	−0.93	−0.154
	64	F		beta	0.22	0.048
	31	F		beta	−1.1	−0.219
Wang 2018 [19]	865	both	GDS-15	d	−0.101	−0.050
Laredo-Aguilera 2019 [27]	16	F	GDS-15	r	−0.379	−0.379
Ling 2010 [25]	484	both	GDS-15	r_eq_	−0.148	−0.148
Stessman 2017 [31]	327	both	BSI	OR	0.668	−0.110
	384			OR	1.009	0.003
	1187			OR	0.545	−0.165
	406			OR	0.320	−0.300
Seino 2013 [29]	340	F	CES-D	OR	0.357	−0.273
Olgun 2019 [34]	144	M	GDS-15	phi	0.192	−0.293
	137	F		phi	0.414	−0.264
Chen 2017 [26]	1081	both	HADS	r_eq_	−0.125	−0.125
Kim 2016 [28]	983	both	GDS-15	beta std.	−0.180	−0.180
Brooks 2018 [20]	3421	both	PHQ-9	beta	−0.190	−0.041
Brown 2012 [21]	854	both	CES-D	d	−0.454	−0.221
Hamer 2011 [23]	210	M	GDS-15	d	−0.388	−0.190
	469	F		d	−0.198	−0.099
Hamer 2015 [12]	3862	both	CES-D	d	−0.341	−0.168
Han 2019 [24]	1056	both	PHQ-9	OR	0.541	−0.167
				OR	0.735	−0.084
Holmquist 2017 [33]	490	both	GDS-15	d	−1.315	−0.549

GDS—Geriatric Depression Scale, PHQ-9—Patient Health Questionnaire-9 (PHQ-9), BSI—The Brief Symptom Inventory, CES-D—Center for Epidemiological Studies Depression Scale, HADS—Hospital Anxiety and Depression Scale. M—male, F—female. OR—odds ratio, beta—regression coefficient, beta stand. standardized regression coefficient, d—Cohen’s effect size, r—Pearson’s correlation coefficient, phi—Yule’s Coefficient of Association oraz r_eq_—Effect size indicator representing the strength of the relationship between the handgrip strength and depression.

## Data Availability

The data analyzed during the study are available from the authors on reasonable request.

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
