# Peer review of "Correlation between Handgrip Strength and Depression in Older Adults—A Systematic Review and a Meta-Analysis"

_ijerph, 2021, doi:10.3390/ijerph18094823_

Round 1

Reviewer 1 Report

It has been a pleasure to have reviewed the work entitled “Association between handgrip strength and depression in older adults - a systematic review and a meta-analysis”. This manuscript evaluated the relationship between depression and strength among older populations using the available literature. It is a very well written and clear document that is easy to follow. The study has been designed well, and the procedures of systematic review appropriately. The findings presented here will be of interest to researchers and health professionals.

I could find very few problems with this manuscript; the authors should be commended for preparing such a high-quality paper. The results are very well presented, and the accompanying figures and tables are easy to understand. The discussion is very well done, and the bibliography is well updated and well selected.

Minor comments are listed below:

Introduction

While I understand that the new criteria for diagnosing sarcopenia both include lack of muscle mass and HGS, it would be interesting to mention the difference between sarcopenia and dynapenia. I am not sure that all studies in the literature follow a correct standardization regarding the classification of sarcopenia.

Search strategy

1)    The combinations of the keywords may be insufficient. It seems the authors used the combinations of four terms, (‘handgrip strength’ AND ‘depression’ AND ‘older’ OR ‘elderly’). The authors should try to include as many as possible the variates of these terms. However, here the authors have to select search terms carefully based on their objective and definition of "depression". 
2)    Not all databases were used. For instance: SPORT Discus (EBSCO), Medline, Embase, Emcare, Scopus, The Cochrane Library
3)    Why wasn't a manual search also used? 
4)    Was a cross-reference search performed after identifying the original studies?
5)    Were participants considered seniors over 60 or 65? Please specify the identification threshold in the introduction. A suitable reference is also needed.

Table 1. Authors should clearly indicate whether the studies were observational or experimental/interventional in the table. Reorganizing the table based on the objectives of the studies makes it much easier to read and understand the reviewed studies.

Figure 1. Please remove red error bars from the Figure.

I have also checked the Author's instructions and find that supplementary files are permitted (check under supplementary material). Hence, the PRISMA checklist could be provided.

Author Response

Dear Editor,

Thank you very much for your letter regarding our manuscript.

We also would like to thank the Reviewer for their comments, which helped us to improve the manuscript. The paper has been revised by a native speaker and by us to improve the text and make it clear and easier to follow. Below we have answered the Reviewer’s comments. Additionally, we have attached the new version of the manuscript.

We hope that in its amended form the article is now suitable for publication.

Yours sincerely,

The authors

To Reviewer 2

It has been a pleasure to have reviewed the work entitled “Association between handgrip strength and depression in older adults - a systematic review and a meta-analysis”. This manuscript evaluated the relationship between depression and strength among older populations using the available literature. It is a very well written and clear document that is easy to follow. The study has been designed well, and the procedures of systematic review appropriately. The findings presented here will be of interest to researchers and health professionals.

I could find very few problems with this manuscript; the authors should be commended for preparing such a high-quality paper. The results are very well presented, and the accompanying figures and tables are easy to understand. The discussion is very well done, and the bibliography is well updated and well selected.

Response: Thank you.

Minor comments are listed below:

Introduction

While I understand that the new criteria for diagnosing sarcopenia both include lack of muscle mass and HGS, it would be interesting to mention the difference between sarcopenia and dynapenia. I am not sure that all studies in the literature follow a correct standardization regarding the classification of sarcopenia.

Response: Has been supplemented in the text

Search strategy

1)    The combinations of the keywords may be insufficient. It seems the authors used the combinations of four terms, (‘handgrip strength’ AND ‘depression’ AND ‘older’ OR ‘elderly’). The authors should try to include as many as possible the variates of these terms. However, here the authors have to select search terms carefully based on their objective and definition of "depression".

Response: During the analysis, the authors checked that all qualified articles contained confirmed information about depression in the studied group of patients.

2)    Not all databases were used. For instance: SPORT Discus (EBSCO), Medline, Embase, Emcare, Scopus, The Cochrane Library

Response: We are very grateful for that comment. We performed searches in PUBMED and Web of Science and Science Direct as databases which we are most familiar with. Also because their query forms are similar, we were able to use the same research equation. We were following the methodology used by other authors.

Examples of systematic reviews carried out in an even smaller number of two databases are:

Cibeira, N.; Lorenzo-López, L.; Maseda, A.; López-López, R.; Moreno-Peral, P.; Millán-Calenti, J. C. (2020): Realidad virtual como herramienta de prevención, diagnóstico y tratamiento del deterioro cognitivo en personas mayores: revisión sistemática. w: Revista de neurologia 71 (6), s. 205–212. DOI: 10.33588/rn.7106.2020258.

Cieślik, Błażej; Mazurek, Justyna; Rutkowski, Sebastian; Kiper, Paweł; Turolla, Andrea; Szczepańska-Gieracha, Joanna (2020): Virtual reality in psychiatric disorders: A systematic review of reviews. w: Complementary therapies in medicine 52, s. 102480. DOI: 10.1016/j.ctim.2020.102480.

Freitag, Fernanda; Brucki, Sonia Maria Dozzi; Barbosa, Alessandra Ferreira; Chen, Janini; Souza, Carolina de Oliveira; Valente, Débora Francato i wsp. (2019): Is virtual reality beneficial for dual-task gait training in patients with Parkinson's disease? A systematic review. w: Dementia & neuropsychologia 13 (3), s. 259–267. DOI: 10.1590/1980-57642018dn13-030002.

Montana, Jessica Isbely; Tuena, Cosimo; Serino, Silvia; Cipresso, Pietro; Riva, Giuseppe (2019): Neurorehabilitation of Spatial Memory Using Virtual Environments: A Systematic Review. w: Journal of clinical medicine 8 (10). DOI: 10.3390/jcm8101516.

Segawa, Tomoyuki; Baudry, Thomas; Bourla, Alexis; Blanc, Jean-Victor; Peretti, Charles-Siegfried; Mouchabac, Stephane; Ferreri, Florian (2019): Virtual Reality (VR) in Assessment and Treatment of Addictive Disorders: A Systematic Review. w: Frontiers in neuroscience 13, s. 1409. DOI: 10.3389/fnins.2019.01409.

Triegaardt, Joseph; Han, Thang S.; Sada, Charif; Sharma, Sapna; Sharma, Pankaj (2020): The role of virtual reality on outcomes in rehabilitation of Parkinson's disease: meta-analysis and systematic review in 1031 participants. w: Neurological sciences : official journal of the Italian Neurological Society and of the Italian Society of Clinical Neurophysiology 41 (3), s. 529–536. DOI: 10.1007/s10072-019-04144-3.

Tuena, Cosimo; Serino, Silvia; Dutriaux, Léo; Riva, Giuseppe; Piolino, Pascale (2019): Virtual Enactment Effect on Memory in Young and Aged Populations: a Systematic Review. w: Journal of clinical medicine 8 (5). DOI: 10.3390/jcm8050620

There are also available reviews with only one database searched such as:

Freeman, D.; Reeve, S.; Robinson, A.; Ehlers, A.; Clark, D.; Spanlang, B.; Slater, M. (2017): Virtual reality in the assessment, understanding, and treatment of mental health disorders. w: Psychological medicine 47 (14), s. 2393–2400. DOI: 10.1017/S003329171700040X.

Marín-Morales, Javier; Llinares, Carmen; Guixeres, Jaime; Alcañiz, Mariano (2020): Emotion Recognition in Immersive Virtual Reality: From Statistics to Affective Computing. w: Sensors (Basel, Switzerland) 20 (18). DOI: 10.3390/s20185163.

Rutkowski, Sebastan; Kiper, Pawel; Cacciante, Luisa; Cieślik, Błażej; Mazurek, Justyna; Turolla, Andrea; Szczepańska-Gieracha, Joanna (2020): Use of virtual reality-based training in different fields of rehabilitation: A systematic review and meta-analysis. w: Journal of rehabilitation medicine 52 (11), jrm00121. DOI: 10.2340/16501977-2755.

Zitzmann, Nicola U.; Matthisson, Lea; Ohla, Harald; Joda, Tim (2020): Digital Undergraduate Education in Dentistry: A Systematic Review. w: International journal of environmental research and public health 17 (9). DOI: 10.3390/ijerph17093269

However, we strongly agree with the Reviewer, that higher number of searched databases will lower the risk of missing important articles. Thank you for that advice we will surely perform more extend search during next systematic reviews

3)    Why wasn't a manual search also used?

Response: Also, the reference sections of the included articles were manually inspected to identify additional records.

4)    Was a cross-reference search performed after identifying the original studies?

Response: Yes

5)    Were participants considered seniors over 60 or 65? Please specify the identification threshold in the introduction. A suitable reference is also needed.

Response:  People over 60 are in the inclusion criteria. Taking this criterion we are informed on the WHO website. Data on older people are from the age of 60, e.g. here  https://www.who.int/news-room/fact-sheets/detail/ageing-and-health,

Table 1. Authors should clearly indicate whether the studies were observational or experimental/interventional in the table. Reorganizing the table based on the objectives of the studies makes it much easier to read and understand the reviewed studies.

Response: All included studies are observational

Figure 1. Please remove red error bars from the Figure.

Response: The figure was changed

I have also checked the Author's instructions and find that supplementary files are permitted (check under supplementary material). Hence, the PRISMA checklist could be provided.

Response: The PRISMA checklist has been included

Reviewer 2 Report

Dear authors,

The study is very interesting and easy to read and to understand. But I have some comments and important questions:

  • The abstract has some words separated in two parts by a "-". You should correct that.
  • You are assessing the correlation between handgrip strength and depression. Thus, please change the title and avoid using the word association in the abstract and the text (correlation does not mean casualty)
  • At the beggining of the methods sections you say "were established and included in the protocol" but you do not include or cite any systematic review protocol.
  • I would delete the sentence in line 75-76 (As the study was retrospective in nature, the approval of the Ethics Committee was not necessary). The Ethics Committee is not necessary because it is a systematic review, and others researchers know that.
  • There must be a mistake in your search equation because I have used in Pubmed and it showed more than 2 million results. Please write it exactly as you write in the database.
  • A "data analysis" subsection title must be included in the methods section before the last paragraph.
  • How did you calculate the correlation of each study based on beta coefficients and other values? Which value did you use for the meta-analysis calculation?. Please indicate it in the data analysis.
  • Figure 2 forestplot in quite hard to read, the letters and numbers are mixed because they are quite toghether.
  • A limitations sections must be included in the discussion.

Author Response

Dear Editor,

Thank you very much for your letter regarding our manuscript.

We also would like to thank the Reviewer for their comments, which helped us to improve the manuscript. The paper has been revised by a native speaker and by us to improve the text and make it clear and easier to follow. Below we have answered the Reviewer’s comments. Additionally, we have attached the new version of the manuscript.

We hope that in its amended form the article is now suitable for publication.

Yours sincerely,

The authors

To Reviewer :

The study is very interesting and easy to read and to understand. But I have some comments and important questions:

The abstract has some words separated in two parts by a "-". You should correct that.

Response: Thank you for the information, everything has been corrected in the text.

You are assessing the correlation between handgrip strength and depression. Thus, please change the title and avoid using the word association in the abstract and the text (correlation does not mean casualty)

Response: As suggested by the reviewer, association will be changed to correlation.

At the beggining of the methods sections you say "were established and included in the protocol" but you do not include or cite any systematic review protocol.

Response: The PRISMA checklist has been included

I would delete the sentence in line 75-76 (As the study was retrospective in nature, the approval of the Ethics Committee was not necessary). The Ethics Committee is not necessary because it is a systematic review, and others researchers know that.

Response: As suggested, the selected sentence has been removed.

There must be a mistake in your search equation because I have used in Pubmed and it showed more than 2 million results. Please write it exactly as you write in the database.

Response: In our research we had exactly the following search words (hand grip strength) AND (depression) AND (older OR elderly), so we found a different number of searches

A "data analysis" subsection title must be included in the methods section before the last paragraph.

Response: As suggested by the reviewer, it has been changed

How did you calculate the correlation of each study based on beta coefficients and other values? Which value did you use for the meta-analysis calculation?. Please indicate it in the data analysis.

Response: Has been supplemented in the text

Figure 2 forestplot in quite hard to read, the letters and numbers are mixed because they are quite toghether.

Response: The figure has been changed

A limitations sections must be included in the discussion.

Response: It has been completed

Round 2

Reviewer 2 Report

Dear authors,

Thank you for addressing my changes.

Kind regards